# Impact of the veterinary feed directive on Ohio cattle operations

Mary Ellen Dillon[1¤]*, Douglas Jackson-Smith[2]

**1** Sustainability Program, Harvard Extension School, Harvard University, Cambridge, MA, United States of America, **2** School of Environment and Natural Resources, The Ohio State University, Wooster, Ohio, United States of America

¤ Current address: Biology Department, University of Dayton, Dayton, OH, United States of America
* mdillon1@udayton.edu

## Abstract

Widespread use of antibiotics in U.S. livestock operations has been identified as a potential contributor to the rising rates of antibiotic-resistant bacterial infections. In response, the U.S. Food and Drug Administration (FDA) issued new rules in January 2017. GFI (Guide for Industry) #213 banned use of antibiotics for growth promotion and required veterinarian permission, via a revised Veterinary Feed Directive (VFD), to deliver antibiotics through feed. Many stakeholders expressed pre-implementation concerns regarding the rules' potential adverse effects on production and profitability. Our study employed qualitative and quantitative methods to investigate how implementation of GFI #213/VFD impacted Ohio cattle operations. We interviewed over fifty cattle farmers and eight large animal veterinarians to document changes in farm antibiotic use, management practices, and profitability. We also examined published government data for possible effects on overall meat production at the state and national levels. We found that the great majority of Ohio farmers reported little difficulty in complying with the VFD with minimal adverse impacts. Farm responses to the feed directive varied with operation size, type (beef or dairy), and whether producers had previously used fed antibiotics. The most commonly reported changes, by both producers and veterinarians, were more veterinary-client interactions, more paperwork/record-keeping, and decreased use of fed antibiotics. All veterinarians, many beef operators, but no dairy operators reported perceiving the VFD as beneficial; however, dairy operations reported less difficulty with compliance due to established working relationships with veterinarians. We found no evidence that the rules impacted the trajectory of state or national livestock output. In conclusion, GFI #213 was reported as not burdensome enough to prevent compliance, but inconvenient enough to incentivize reduced use of fed antibiotics (when previously used) without significant adverse effects, consistent with its goal of promoting judicious use of medically important antibiotics in order to preserve their effectiveness.

## Introduction

The use of antibiotics has not only saved millions of human lives, but has also extended life span and enhanced quality of life [1]. Thus, the increasing frequency of bacterial infections

**Data Availability Statement:** Because of constraints associated with Institutional Review Board requirements, specifically the IRB human subjects protocol restrictions used for this study, prevent us from releasing any data that could lead to the identification of study participants. The topic

of antibiotic resistance is a very sensitive one in Ohio, and we were only able to secure access to research subjects by promising complete confidentiality regarding their participation and their answers. The narrative comments in interview transcripts and detailed information about each farm embedded in comments could be used to reveal the identity of the respondents, so disclosure of that material is prohibited by our approved IRB protocol. We believe that stripping identifying information from the qualitative dataset will render it unusable for replication purposes. In our supplement file, S4_File.slsx, We have provided a limited quantitative farmer and veterinarian dataset that includes answers to core closed-ended questions about how respondents were impacted by the VFD, as well as basic information about farm type (Dairy vs. Beef) for the producer survey.

**Funding:** D.J. received funding for the research reported here from a seed grant from The Ohio State University's Initiative for Food and AgriCultural Transformation (InFACT), a Discovery Themes program (discovery.osu.edu/infact). The funder played no role in the study design, data collection or analysis, decision to publish, or preparation of the manuscript. There was no additional external funding received for this study.

**Competing interests:** The authors have declared that no competing interests exist.

resistant to antibiotics is a great global health concern [2]. Unfortunately, the more an antibiotic is used, the greater the probability that bacterial populations evolve resistance to that antibiotic due to natural selection. Thus, the American Medical Association has urged US physicians to use antibiotics judiciously, that is, sparingly and only when appropriate, to combat the growing incidence of resistance [3].

Interestingly, the bulk of medically important antibiotics (MIA) in the US have been sold, not for human use, but for use in animal husbandry to treat, control, and prevent disease, as well as for growth promotion [3]. Thus, zoonotic bacteria can become resistant in food-animals and then be transferred to humans. On January 1, 2017, the US Food and Drug Administration (FDA) responded to growing public health concerns by implementing GFI (Guide for Industry) #213 which restricted the use of antibiotics in livestock production. A key component of this mandate was a revision of the Veterinary Feed Directive (VFD), which requires veterinarian oversight and permission to administer antibiotics to livestock through food [4].

The FDA recognized that these new policies would result in "significant" practical impact for both animal producers and veterinarians [4]. In public forums occurring prior to the implementation of the revised VFD, some producers and livestock organizations expressed anxiety about the potential adverse effects the rule may have on farm profitability and livestock production. To better understand the validity of these concerns and assess the actual impact on stakeholders, we used qualitative and quantitative methods to answer the research question: How has the implementation of the revised VFD impacted Ohio cattle operations in terms of operator outlook, management practices, and meat production levels?

## Antibiotic use in livestock production

Certain bacteria not only cause illness in humans but also in livestock animals (poultry, swine, cattle) commonly raised for food. Thus, agriculture has also benefited by the discovery of antibiotics, gaining an important tool to treat bacterial infections. For example, antibiotics are commonly used to treat mastitis in lactating cows and respiratory disease in calves [5]. Today, antibiotics are used not only to treat individual animals, but also to control herd or flock disease outbreaks and to prevent disease by administration through food or water.

A more controversial use of antibiotics quickly developed in the 1940s when the poultry industry discovered that the use of tetracycline increased the rate of weight gain [6]. Although the reasons for this increased weight gain were (and remain) poorly understood, this practice quickly spread throughout the livestock sector. In the 1950s, the accelerated use of antibiotics in livestock production was concurrent with the rise of intensive feeding operations [7]. High animal densities facilitated the spread of infectious pathogens which antibiotics prophylactically countered.

In the US, farmers have been able to buy most antibiotics "over the counter," or without a prescription from a medical professional, a practice not permitted with respect to human health. In addition, until recently US farmers have been able to purchase antibiotics for "growth promotion" or "prophylactically" without veterinarian oversight. In contrast, the European Union has banned the use of antibiotics in agriculture as growth promoters since January 1, 2006 [8]. The World Health Organization (WHO) has recommended that agriculture cease using antibiotics in livestock for either growth or disease prevention [9].

Much variation exists in animal husbandry practices among US food animal sectors. The cattle industry was responsible for 43% of the MIAs consumed by US livestock in 2016—the first year of mandatory reporting by animal species—compared to 6% for the chicken industry [10]. In contrast to the beef industry, the chicken industry has responded more quickly to consumer demand to offer more NAE or "No Antibiotics Ever" product lines. Chicken sold as

NAE increased from 4% in 2013 to 13% in 2015 to over 50% in 2018 [11]. Interestingly, turkey operations, which comprise far less biomass production than chickens, account for roughly 10% of antibiotic consumption by livestock in years 2016–18 [10].

In order to assist the FDA in its analysis of agricultural antibiotic use and safety concerns, US animal drug sponsors have been required, since 2009, to report their sales of antibiotics to the FDA. Likewise, the FDA is required by law to publish a summary report by December 31 of the following year. Beginning in 2016, drug sponsors have also been required to report antibiotic distribution by animal species (i.e. cattle, swine, chicken, turkey). In the first three years of mandatory reporting, antibiotics sold for livestock consumption steadily increased, with the total amount of antibiotics sold in 2012 showing a 23% increase from the 2009 levels [10].

During this period, only two tools existed to regulate how antibiotics were being used by the food-animal sector, namely prescriptions (typically for administration through water) and a veterinary feed directive (VFD), that is, a written order from a veterinarian to administer antibiotics through feed [12]. The latter was rarely used, however, because most antibiotics were available over the counter.

In 2012, the FDA issued GFI (Guide for Industry) #209 entitled, *The Judicious Use of Medically Important Antimicrobial Drugs in Food-Producing Animals*, in which it outlined its concerns that antibiotics were losing their effectiveness against bacterial infections due to injudicious practices in both human and food-producing animal populations [13]. It also stated its thinking that using antibiotics for "production" or "feed efficiency" purposes were injudicious and expressed concerns over resistant bacteria being transmitted to humans. A year and a half later, the agency issued GFI #213 which established a timeline to implement the goals of GFI #209 and requested voluntary compliance by pharmaceutical companies and livestock producers [4]. A key part of GFI #213 was a revision of the veterinary feed directive (VFD) rules, which had been described as overly "burdensome" by producers [12]. The FDA published the VFD final rule in 2015, setting the stage for full implementation of these policies [12].

On January 1, 2017, full enforcement of GFI #213 began, which mandated new restrictions on the use of antibiotics in livestock operations. Most medically important antibiotics (MIA) intended for use in feed or water were no longer permitted to be sold "over the counter" but required oversight by a licensed veterinarian who had an established professional relationship with the client (farmer), known as the VCPR (Veterinarian Client Patient Relationship). Under GFI #213, most antibiotics administered through water required a prescription. Farm operators wishing to mix antibiotics into livestock feed–the most common route of administration–were required to obtain a written Veterinary Feed Directive (VFD). The use of antibiotics prophylactically (as preventative) was still permitted; however, the rule made the practice of using antibiotics for purposes of growth, "production", and/or "feed efficiency" illegal.

## GFI #213 and production concerns

Traditionally US meat-producers have opposed a ban on the use of antibiotics as growth promoters, arguing that animal health would suffer, meat production would decrease, and the industry would suffer economic harm [14]. These concerns were visible in the public debates over the GFI #213, where producer advocacy groups, such as the America Farm Bureau Federation, argued that restricting access to antibiotics posed a risk to animal health and the ability to produce safe product [15].

Meanwhile, evidence from the European Union (EU) suggested that these concerns may be overstated. Denmark was one of the first EU countries to limit such antibiotic use in the 1990s followed by a complete ban on antibiotics as growth promoters in 2000. Since the ban, poultry

production has increased slightly while pork production has significantly increased [16]. These production changes corresponded with reductions of antibiotic use by 90% and 51%, in poultry and pork husbandry respectively [17].

Few studies exist that examine the preliminary ground-level impact of the VFD in the U.S. on livestock operations. After the first draft of the rule was proposed, Lee et al. (2017) surveyed beef producers across the US and found that over 80% of producers were familiar with the rule and expressed a mix of positive (27%), negative (41%), or mixed/indifferent (33%) opinions about its impacts [18]. Ekakoro, Caldwell, Strand, and Okafor investigated the impact of the VFD on Tennessee cattle producers during its first year of implementation. They conducted focus groups and distributed a survey to a large sample of beef and dairy producers in Tennessee to gather information about the impacts of the rule on their operations. They found that many producers had a negative perception of the rule, including perceptions that VFD was a top-down policy, limited their options to address animal health problems, and would likely cause economic losses (particularly for small producers) [19]. Roughly 40% of their respondents believed that VFD would lead to increased use of injectable antibiotics, perhaps compensating for some of the reductions in fed antibiotics [19]. That said, most respondents saw merit in the rule; for example, a majority of beef producers (55%) agreed that the VFD was somewhat or very useful. A significant minority of both beef and dairy operators also reported increasing their interactions with veterinarians as a result of the VFD [19].

In a separate paper, Ekakoro et al. reported that use of antibiotics in general varied by type of cattle operation (beef or dairy; size) and was influenced by both market pressures and interactions with veterinarians [20]. They documented an increase since the late 2000s in the proportion of producers keeping records on antibiotic use and a reduction in the off-label use of antibiotics; however, they did not link this to GFI #213 or the VFD [20].

Rademacher, Pudenz, and Schultz surveyed swine veterinarians before and after the first year of VFD implementation [21]. They found that because of the VFD, veterinarians were having more conversations with their swine producer clients regarding the judicious use of antibiotics [21]. The most commonly reported changes in operation management practices included increased vaccinations (81%), increased nutritional feed additives or supplements (57%), and modified biosecurity (49%) [21]. The veterinarians also reported the perception that antibiotic use by their clients had substantially decreased [21]. FDA Summary Reports published since confirm their perceptions [10].

## Ohio as a representative agricultural state

Ohio is a major agricultural state with a diverse livestock sector and serves as an interesting place to investigate the impacts on and responses by livestock producers to GFI #213. According to the 2017 USDA Census, Ohio is home to over 25,000 farms that raise cattle, over 10,000 farms who raise poultry, and roughly 3,500 farms that raise swine [22]. Swine and poultry production in Ohio are highly integrated, meaning that the majority of the animals are owned by an "integrator," a company that contracts farmers to raise the livestock animals in accordance with strict company mandated protocols. Access to these operations can be difficult due to biosecurity concerns. In contrast, the Ohio beef and dairy farms tend to be individually operated and thus are more diverse in their management practices. With its diverse farm management styles and scales of operation, Ohio cattle industry provides an excellent opportunity to explore stakeholder perspectives of the VFD and its impact on a local level.

The goal of GFI #213 was to promote judicious use of antibiotics in the livestock sector, however, a mandate is meaningless if it is too difficult or cumbersome to implement. In keeping with the "One Health" approach [23], our research sought multiple perspectives and had

two foci. The first objective of this study was to explore the perceptions of Ohio cattle farmers (both beef and dairy) and Ohio large animal veterinarians regarding the VFD and its impact. Our second objective was to test the conventional hypothesis that the mandated restrictions on antibiotic use would negatively affect production. To our knowledge, this is the first study to explore the perceptions of the VFD by primary stakeholders in Ohio livestock production.

## Materials and methods

In keeping with the "One Health" philosophy, our goal was to investigate the impact of the Veterinary Feed Directive (VFD) from multiple perspectives. We chose to use a mixed methods approach that entailed the collection of primary qualitative data (stakeholder interviews) as well as the analysis of publicly available secondary data. The human subject research was reviewed and determined exempt by the Ohio State University Institutional Review Board on December 18, 2018 (study #2018E0880).

### Sampling and recruitment

To assess the farm level impacts of the VFD, we conducted interviews with Ohio cattle producers (both beef and dairy) and veterinarians. The interviews were conducted in two phases. The first was in conjunction with a study by researchers from The Ohio State University (OSU), *Assessing the Role of Cattle Production Systems and Landscape Characteristics in Antimicrobial Resistance Profiles using a Participatory Epidemiology Approach* (R. Garabed PI). The team was conducting interviews with representative dairy and beef producers as part of a broader study of antibiotic management and the ways that landscape and producer diversity impacts the prevalence of antibiotic-resistant bacteria on working farms. The team agreed to incorporate additional questions regarding the impact of the VFD in their interview instrument (S1 File). Questions included both open- and closed-ended items. This phase included producers from three Ohio counties (Mercer, Wayne, and Muskingum) that represent different levels of landscape and farm diversity. Building upon the results of Phase I, we revised the interview instruments to gain a deeper and more nuanced understanding of the ground-level impact of the VFD and recruited additional cattle farm operators and large animal veterinarians (S2 and S3 Files). Phase II focused mainly on cattle farmers operating in Western Ohio (Adams, Clark, Highland, Mercer, and Montgomery counties) and veterinarians throughout the state.

In both phases, we initially identified potential interviewees through meetings and professional contacts with local OSU county extension educators, Soil and Water Conservation Service staff, leaders of the Ohio Farm Bureau and Ohio Cattleman's Association, the Ohio State Veterinarian's office, and publicly advertised cattle operations. At the end of each interview, respondents were asked to volunteer names (and contact information) for other livestock operations in their area who might be able to provide additional perspectives. Through this snowball sampling approach, we were able to identify and recruit a number of additional farms. Farmers and veterinarians were contacted by email and phone to gauge their interest in participating. While this approach was not designed to achieve a true random sample, we attempted to minimize sampling bias by purposively selecting participants who represented all farm sizes as classified by the USDA 2017 census, as well by soliciting both beef and dairy farm operators. Across the two phases of the study, we received useable interview data from 54 cattle farms, including 39 beef and 15 dairy operations (Table 1). Compared to census data, our final sample underrepresents the smallest cattle farms (< 50 head) but includes the full range of USDA farm size classes (Fig 1).

**Table 1. Total cattle farm operators interviewed.**

|  | Beef | Dairy | Total |
|---|---|---|---|
| Phase 1 | 23 | 12 | 35 |
| Phase 2 | 16 | 3 | 19 |
| Total | 39 | 15 | **54** |

## Data collection and analysis

The interviews were semi-structured and occurred at a time and in a place of the participant's convenience, either in person or by phone between May 2019 –January 2020. The Phase I interviews were conducted in pairs with one person asking questions and another taking notes. One or both of the authors were present at all but six of the Phase I interviews; the remaining six were facilitated by a third lead OSU interviewer. The lead author conducted 100% of the Phase II interviews. All interviews were recorded (with the participants' permission), transcribed, and treated confidentially. Both authors had access to the recordings and transcripts from both phases of interviews.

The interview instruments contained a mixture of open-ended questions (exploring antibiotic use and philosophy, in addition to farm demographics and landscape characteristics) as well as a series of quantitative Likert scale questions which focused on the impact of the VFD. One week into Phase I, a Likert question assessing the VFD impact on the frequency of veterinarian interactions was added. In the Phase II producer instrument, many open-ended questions which did not pertain to the VFD or antibiotic use were deleted, and two additional Likert scale questions (regarding the impact on paperwork and fed antibiotics) were added. A similar parallel instrument was developed for the veterinarians.

The quantitative data from the six Likert scale questions common to both phases of producer interviews were combined. One of the additional questions (use of fed antibiotics) had no Phase I Likert counterpart, but it nearly always arose in answers to open-ended questions about the impacts of the VFD. Using the transcripts from the interviews, we were able to infer

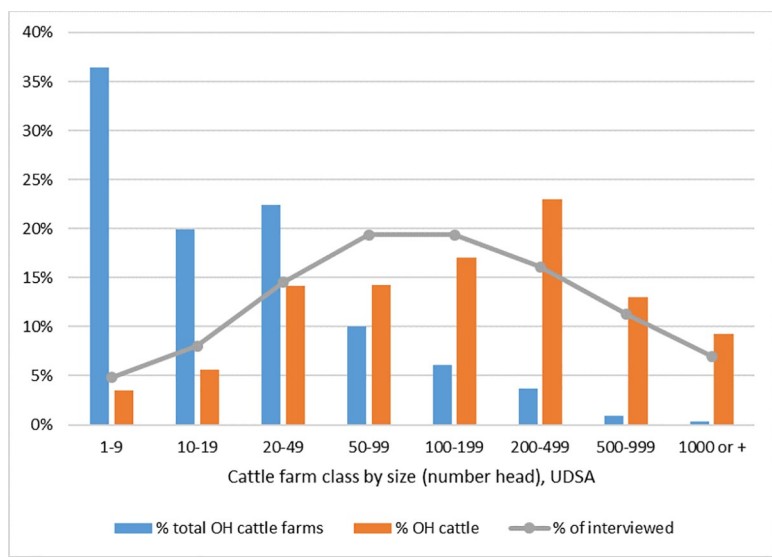

**Fig 1. Size distribution of Ohio cattle farms interviewed for this study.** Percent interviewed as compared to percentage of OH farms by USDA size class and percentage of animals (number head).

whether most Phase I farms had previously used fed antibiotics prior to the VFD, and by how much (or if) their use of fed antibiotics had changed since the implementation of the rule. We used descriptive statistics to analyze the frequency of quantitative (close-ended) questions.

Transcripts of the interviews were systematically analyzed and coded for emergent themes and patterns using established methods of qualitative data analysis [24, 25]. Specifically, both authors independently identified components of answers to each question that could be coded into distinct themes, and collaboratively refined the criteria for code assignment until both authors converged on similar coding assignments. Both the qualitative and quantitative results were compared and contrasted across farm types and stakeholder groups.

To examine production trends, we compiled state and national-level data on cattle production between 2009–2018 from annual reports issued by the USDA National Agricultural Statistics Service (NASS) 'QuickStats' tool [26]. Ohio ten-year trend lines were graphed with the corresponding national data for context. We examined the trend lines for obvious deflection points coinciding with the implementation of GFI #213 to determine if additional statistical analysis was warranted.

## Results

One of the most important, and somewhat surprising, finding of the interviews was that Ohio cattle producers and veterinarians reported little difficulty in complying with the Veterinary Feed Directive (VFD). Although much concern was initially expressed in public forums across stakeholder (farmer and veterinarian) sectors, the great majority of interviewees reported that, after initial adjustments, compliance was fairly straightforward and without major adverse impact on farm profitability or production. In fact, over 90% of all cattle producers reported either "no change" in operation practices or said that compliance was "not difficult" or even "easy." Less than 5% reported significant (though still minor) challenges and none reported major difficulty in complying with the VFD. Despite this general trend, interesting variations existed among and between the stakeholder groups.

### The impact of the VFD on cattle producers

Table 2 reports the percent of farmers in our sample who reported changes in various management practices and other on-farm outcomes as a result of the VFD. Most farmers reported no or minimal impacts, with 90% of responses to most items either "no change," "decreased a

**Table 2. Changes due to VFD reported by interviewed Ohio cattle operations.**

| Type of Change | Mean score | N | Decreased a lot | Decreased a little | No change | Increased a little | Increased a lot |
|---|---|---|---|---|---|---|---|
| | | | | | *percent of respondents* | | |
| *(Phase 1 and 2 combined)* | | | | | | | |
| Total use of antibiotics | 2.9 | 52 | 2 | 12 | 77 | 10 | 0 |
| Amount of fed antibiotics used | 2.5 | 53 | 19 | 9 | 72 | 0 | 0 |
| Use of vaccines | 3.1 | 50 | 0 | 0 | 96 | 2 | 2 |
| Use of nutritional supplements | 3.2 | 50 | 0 | 0 | 88 | 6 | 6 |
| Health of livestock | 3.0 | 53 | 0 | 4 | 94 | 2 | 0 |
| Profitability of farm enterprise | 3.0 | 52 | 0 | 10 | 85 | 2 | 2 |
| Number of interactions with veterinarian | 3.3 | 47 | 0 | 0 | 77 | 17 | 6 |
| Amount of paperwork | 3.6 | 18 | 0 | 0 | 50 | 39 | 11 |

Mean score refers to responses to 1–5 Likert Scale in which 1 = Decrease a lot, 2 = Decrease a little, 3 = No Change, 4 = Increase a little, 5 = Increase a lot. Not all questions were answered by all participants and one item was only included in Phase 2.

little" or "increased a little." The greatest variations were seen in response to questions regarding the use of fed antibiotics (30% reported a decline), interactions with veterinarians (23% reported an increase), and amount of paperwork (50% reported an increase).

Further analysis revealed that the reported effects of the VFD on Ohio cattle operations varied by three key farm characteristics:

1. Whether or not the farm operator had used fed antibiotics prior to the implementation of the feed directive

2. Type of operation (dairy versus beef)

3. Herd size

**Prior administration of antibiotics through animal feed.** Roughly 44% (24 out of 54) of the cattle operations interviewed had not been using fed antibiotics prior to the VFD. Unsurprisingly, this group reported being generally unaffected by the VFD. Of the roughly 56% (30 out of 54) cattle operators who had previously used fed antibiotics (at least occasionally) in the past, their reported impact varied by type of farm (beef or dairy) and size of operation. Farms that fed antibiotics prior to the VFD were much more likely to experience impacts from the policy change.

*Dairy operations.* The dairy farmers who reported pre-GFI #213 use of fed antibiotics (11 of 15 interviewed), reported having little trouble complying with the VFD. This was largely because dairies generally avoid use of antibiotics with their milking cows because milk must be discarded during a "withhold" period after a cow receives antibiotics. The dairy operators reported that use of fed antibiotics occurred mainly with groups of calves. Additionally, dairy farmers interact more frequently with their veterinarians as part of regular herd health checks, so obtaining a feed directive, when needed, was relatively easy (typically reported as merely an extra step, not something onerous).

Among those who previously fed antibiotics, the most commonly reported impacts of the VFD on dairy operations reflected a significant decrease in the use of fed antibiotics (AB) for calves and a slight increase in the use of nutritional supplements and interactions with their veterinarians (Fig 2).

Interviewee comments provided greater nuance. In response to "How did the VFD change your dairy operation?", operators consistently reported that feeding Aureomycin (tetracycline) "crumbles" to calves during weaning or transitions was less convenient after the VFD, therefore operations tended to reduce the practice. To compensate, some shifted to treating sick calves with electrolytes or individually with injectable antibiotics. Some representative quotes were:

*There's nothing to the milking cows; and the calf side, I would say we use less strictly because of the hassle. (Dairy, large)*

*We were giving quite a bit of antibiotics for scours. We pretty much stopped that and we're doing more of a electrolytes treatments and that kind of thing. They seem to respond pretty well. (Dairy, large)*

*The calves, I would say, I probably do use more injectable. (Dairy, medium)*

They also commented on the need for more frequent interactions with their veterinarians, but suggested it represented a minor increased demand on time:

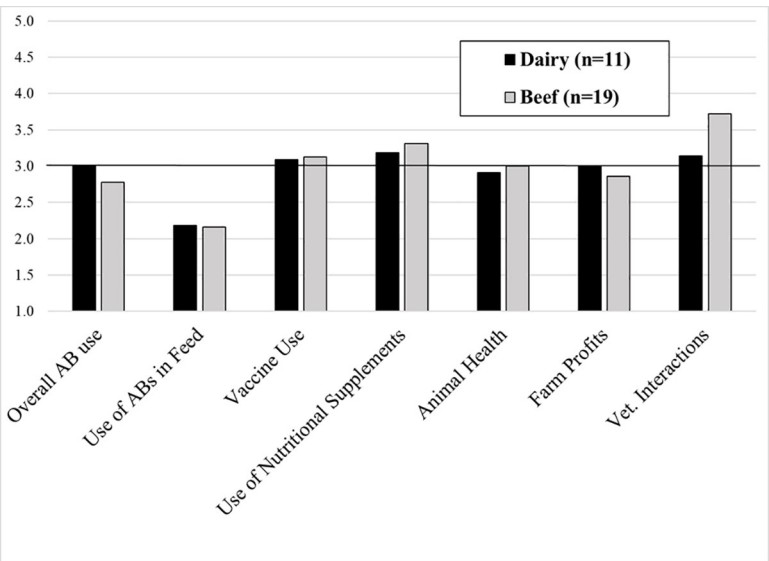

**Fig 2. Impact of VFD on subset of Ohio dairy and beef operations using fed antibiotics prior to the VFD.** Mean score for impact on various aspects of farm management and performance. Responses to 1–5 Likert Scale in which 1 = Decrease a lot, 2 = Decrease a little, 3 = No Change, 4 = Increase a little, 5 = Increase a lot. Horizontal line represents no reported change.

> *You gotta get a VFD from the vet if you want to use an antibiotic in the feed. Other than that, it's not too horrible. (Dairy, medium)*
>
> *Probably the biggest challenges is it created a different set of hoops to go through to be able to get the antibiotic that we've used to put in the feed. (Dairy, large)*
>
> *Just to remember to get the VFD we need when it was up and make sure we had everything in line. (Dairy, medium)*

To summarize, all of the dairies reported little to no difficulty with compliance. Typical comments included "It hasn't been that hard," "Very easy," "Oh, it's been fine. It's a little more hassle," and "Actually, it's more of a nuisance than it is anything else."

*Beef and/or replacement heifer operations.* The non-dairy cattle operations interviewed were primarily cow/calf operations, though our sample also included some finishing operations (both grass-fed and feedlot), some backgrounders (pre-finishing), and a couple of replacement dairy heifer operations. All the cattle were ultimately being raised for beef with the exception of two replacement heifer operations associated with dairies. For simplicity, we will refer to this entire group as 'beef operations.'

The beef producers tended to be more diverse in terms of their approaches to antibiotic use and management. As opposed to the dairy interviews, where most of the operations had regularly used fed antibiotics prior to the VFD, the beef operations reported closer to an even split with 19 of 39 operators reporting at least some use of fed antibiotics prior to the feed directive and 20 of 39 reporting that they never used fed antibiotics.

Of the latter group (never fed antibiotics), beef farmers' reasons were obviously different from dairy farm operators who need be more cognizant of the "withhold" requirement. Among the "no fed antibiotic" beef producers, some held deep convictions regarding the need to use antibiotics as sparingly as possible, often employing strong preventative measures. These included regimented vaccination schedules, good ventilation and regularly cleaned

bedding for animals when in barns, and providing shelter for the herd from wind/sun when outdoors. Many cited the need to reduce stress, and therefore susceptibility to disease, such as one producer who managed his herd by "walking" versus "driving" cattle to new grazing grounds.

Typical comments included:

*Really.,,, I love this barn because the ventilation keeps us from having any air, respiratory problems...But, I mean, just little easy management of stuff and I just don't have much in the way of health problems. (Small, beef)*

*What I've learned is . . . for the cattle-wise, keeping the bedding just clean. Two times a week I scrape the barns and scrape the lots, and make sure that they're in a good clean environment and that really helps. (medium, beef)*

Others in the "no fed" group, usually medium to smaller operations (~ 100 heads or fewer), simply took a more "laissez-faire" approach to management in general, often not even vaccinating their cattle. To summarize, in response to "What is your philosophy or general approach to antibiotic use?" many beef producers articulated a range of personal values that highlighted their preference for judicious or minimal antibiotic use:

*I try to avoid it at all costs...I'm a physician . . . and I've seen within medicine what's happened to our antibiotic resistance. It's just dramatic. And so I was 100% in favor of restricting antibiotic usage. (Beef, medium)*

*It's mostly nonexistent... Never. Hardly ever...Because we don't have a way to catch our cows, so hardly ever anything's done. (Beef, medium)*

Of the nineteen interviewed beef farmers who used fed antibiotics prior to GFI #213, many reported that it was "not difficult" to comply VFD, often citing they already had established a good relationship with their veterinarian. Of those operations affected by the VFD, the most commonly reported changes included reduced use of fed antibiotics (crumbles) for calves, changes in management practices, increased interactions with vets, and increased paperwork (Fig 2; Table 2). Overall, beef producers were also more likely than dairy to report that they increased their interactions with veterinarians.

Before the VFD mandate, antibiotic "crumbles" were readily available over the counter and routinely mixed into livestock food by feed mills. The following comments illustrate the reduced use of fed antibiotics and/or a change in management practices due to the VFD.

*. . . now I would say that we'd probably do that less (use fed antibiotics with calves) and we probably get in individual calves, and give them a shot of antibiotic rather than feeding the whole group. (Beef, large)*

*Now we've combated that, and we've implemented a different vaccination program, so now we're trending (economically), working better than they were before, but we're getting it in a different way. (Beef, large)*

*We would use more injectable antibiotics, as opposed to feed grade antibiotics. (Beef, medium)*

*Size of operation.* The interviews included farms across a wide range of herd sizes (from a handful of cattle to herds of more than 1000). Many informants felt that the larger operations

would be more greatly impacted by the VFD, especially the feedlots, as they were more likely to be using fed antibiotics. This perspective was overall supported by the data. The percentage of beef operators using fed antibiotics generally increased by class of herd size, while fed antibiotics were commonly used on both medium and large dairies (Fig 3A and 3B).

On the other hand, many of the larger farm operators shared their impressions that smaller operations would be more impacted, primarily due to their lack of routine access to a veterinarian. Typical comments included:

> I think the farmers that it hit were the guys that were small, smaller and don't have the vet on the farm all the time. (medium, dairy operator)

> Most of the cattle operations who got any amount of numbers, they follow 'to the T' . . .the Veterinary Directive. The ones where you run into trouble is the guys that have maybe five or 10. They're not big enough to handle any larger amount, so they just have a real battle. (large, beef operator)

> It puts the small guy like me and anybody that can't have the vet on staff, at a major disadvantage. (medium, beef)

These perspectives were supported by the interviews. The two interviewed small (1–19 head) beef operations that had previously used fed antibiotics reported that they had stopped due to the inconvenience/hassle of compliance with the VFD and difficulty in finding an available veterinarian. Perhaps the impact of the VFD on Ohio cattle operations can best be summarized by the following operator comment:

> It's probably not that much of an issue for large or small, but they both have some times when it could be a challenge. (Beef, medium)

## Impact of the VFD on veterinarians

We interviewed eight large animal veterinarians practicing in Ohio. The percentage of their clients who were cattle producers ranged from less than 10% to 80%. They represented a wide range of experience in the profession, from newly graduated from veterinary school to over forty years in practice. Additionally, they represented counties from the most northern to the most southern regions of Ohio. Almost all had established VCPRs (Veterinary Client Patient Relationships), with both beef and dairy producers, collectively representing interactions with hundreds of cattle operations.

The veterinarians interviewed were overwhelming positive about the goal of the VFD as expressed in this representative comment:

> I was pretty positive about it. I thought it was necessary. I feel like there was too many antibiotics used without direction. Things were just dumped in without even having any knowledge of why or what for. And even after it started, we found lots or those situations where people couldn't understand why they couldn't just dump a bag of antibiotics in their feed without some direction.

Despite supporting the goal of the VFD, many were quite worried or concerned about its impact prior to implementation:

> Everybody was kind of freaking out about it when it came out. Nobody really knew what was going on. The listserv for AABP was rife with discussion for months; we just had so much hype

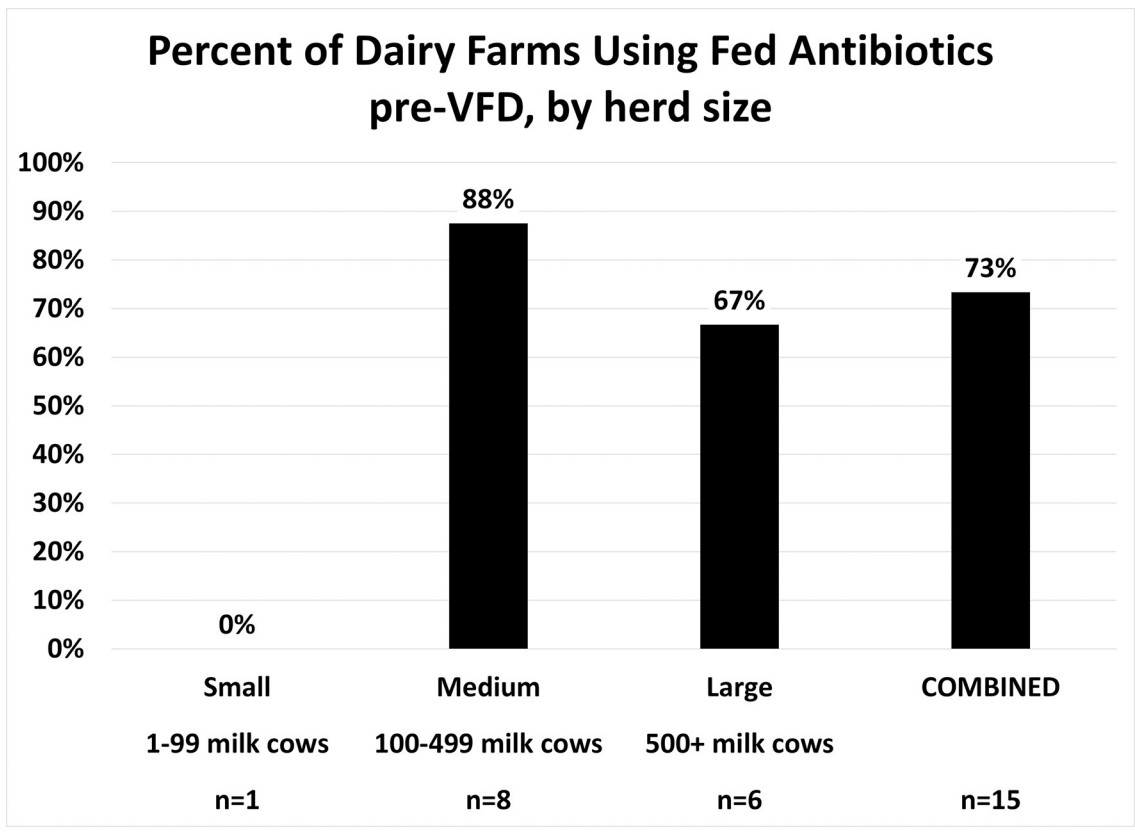

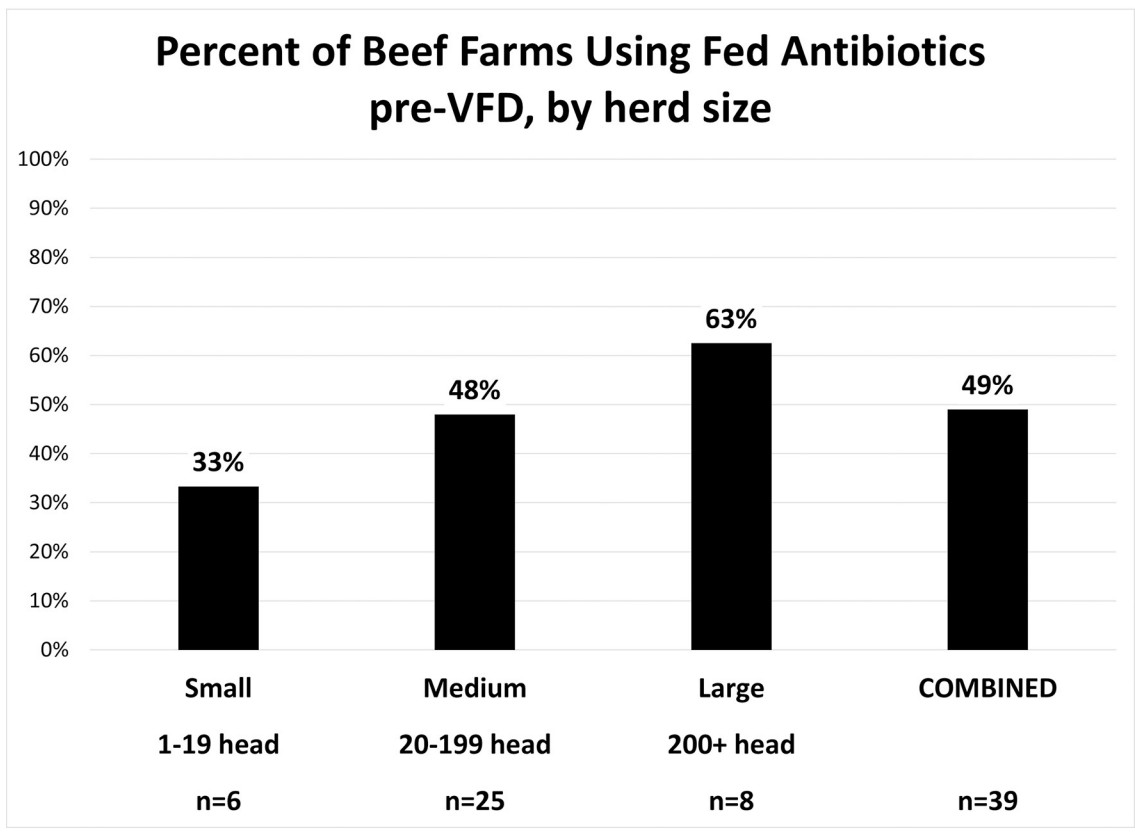

**Fig 3. Use of fed antibiotics by Ohio cattle operations, percent by herd size.** a. Dairy (size of milking herd) b. Beef (total head).

*that this was what we had to do, and we had to do this correctly, and USDA's going to be watching us, and on and on. And about how this is going to be a major change to the way we practice veterinary medicine, and it was not so.*

*So it hadn't been that big a deal. . .people were scared to death it was going to be.*

Consistent with the cattle producers, the veterinarians reported that despite the initial misgivings, compliance with the feed directive was not very difficult. The reported areas of greatest impacts were also consistent with the producer interviews (Table 2; Fig 2): increased number of VCPRs, a decreased amount of fed antibiotics (AB), increased number of client interactions (farm visits), and increased record keeping and paperwork (Fig 4).

The value for "amount of scripts (prescriptions) written for feed" may be underreported. The veterinarians in our earlier interviews were not sure how to respond to this question; the latter participants emphatically stated that this value increased a lot, as they were not writing any feed directives prior to implantation of GFI #213, due to the fact that fed antibiotics were freely available over the counter (OTC).

The increase in client interactions was due, in part, by the requirement that feed directives can only be issued in the context of an established VCPR. In other words, veterinarians were required to visit the farms and have some first-hand knowledge of the operations. Vets reported that this provided them the opportunity to discuss appropriate use of antibiotics with the producers. Representative comments include:

*Oh yes. Yes. There was a significant amount of education about the process, about why we had to do it and that I was just not trying to make more money off of them.*

*I think more than anything, it allowed us to create a discussion on farms to determine why do we need it and what can we do to prevent the need for it.*

Much of the interactions with producer clients centered on communicating a judicious approach to antibiotic use as well as changing producer mindsets that antibiotics should be the first course of action for disease control or prevention. Consistent with many producers,

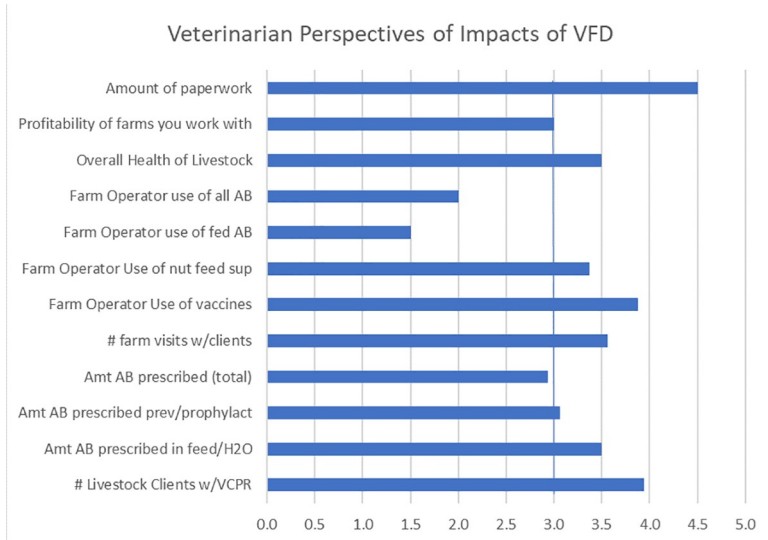

**Fig 4. Mean scores for veterinarian responses describing the impact of VFD.** Likert scale responses (n = 8) in which 1 = Decrease a lot, 2 = Decrease a little, 3 = No Change, 4 = Increase a little, 5 = Increase a lot. Vertical line represents the response of "No Change".

veterinarians emphasized the importance of good preventative management practices which reduce, and at times, even eliminate, the need for antibiotics.

> *I think that they do get overused. I don't tend to use them as a first line. I would rather prevent disease. . .I think they're definitely necessary to save animals and help them, but they should not be used to manage them.*

> *Better vaccination programs, better nutrition, weaning their cattle better, planning better. It's just changed everything.*

> *We have a major slant towards judicious use, which means that we have written protocols for the diseases of interest on our dairies. And we're in the process of updating them right now for our annual update. We are inclined towards injectable for individual animal treatment.*

> *The farmer invested in a positive pressure tube ventilation system. And what we found is he doesn't need to give those antibiotics anymore . . . both feed through antibiotics and injectable antibiotics in that barn.*

Also, consistent with the cattle producers, veterinarians reported increased record- keeping and paperwork as a major impact of the VFD on their practice. They also reported VFD-related frustrations. Despite these reported frustrations, a commonly heard theme from the veterinarians was that the VFD prodded them to become better and more responsible veterinarians. Sample responses were:

> *And then as far as the paperwork goes, it's figuring out the stinking dosages on those things. They can't even make them consistent across the board. It was just irritating, trying to get that. . . Two years into it, we're still figuring it out.*

> *Well, I think the paperwork is the biggest thing.*

> *And so what the VFD has done is targeted that and made the veterinarian responsible and accountable, which is the way it should be.*

I think it's kind of accentuated the VCPR, it kind of gave credibility to the veterinary of record. Veterinarian oversight has become more defined and I think it's good for the industry, not just for me, but I think for the whole industry.

**Additional emergent themes.** In addition to the broad finding that complying with the VFD was relatively easy and more of a nuisance than a major burden, the interviews also revealed that many in the Ohio livestock industry saw the VFD as a good thing. All of the veterinarians expressed this view, as well as about half of the responding beef cattle operators (but none of the dairy farmers). The predominant positive perspectives expressed mirrored the observed quantifiable effects, namely the reduced antibiotic consumption and, interestingly, the increased record-keeping.

> *I saw so much misuse of them. They're talking about treating some animal and. . . I said, 'How much are you treating them with?' 'Oh we're putting about a handful in.' So people like that definitely need to be restricted. Definitely was a lot of misuse. (veterinarian)*

> *Absolutely (needed). And so just from being a nutritionist in the industry, you would be shocked at the yards that. . . that run. . .an antibiotic all the time for no reason at all other than a preventative, and that stuff needed to stop and it did stop. (large, beef)*

*. . . for the feed directive, what I will say has helped me a lot with . . . the record aspect of it, because now that we have to keep records of our antibiotic use, I think that has really benefited our farm. (medium, beef)*

*And really, it does affect us. But it's for the betterment of society. (medium, beef)*

While not systematically asked of all interviewees, differences of opinions were expressed on the use of antibiotics for prevention, either prophylactically (in anticipation of disease onset) or metaphylactally (to control disease, i.e. prevent spread through a herd), much being context specific:

*. . . but when you're dealing with feedlots and you have a bunch of cows moving in and out, metaphylaxis works. It's been shown to work (veterinarian)*

*People complain about CAFOs. . .. As long as they're all in (and) all out, so they're filling up a barn with the same age type animals . . . and you clean and disinfect and you bring another whole group in. If you're continuous flow, where you're bringing in a hundred animals every month. . ., whatever diseases are there will stay there, because you never had a chance to break the cycle. (veterinarian)*

*Well, . . .I don't know what's worse. Having a low dose of Aureomycin in through the winter or having more calves that I have to treat with Resflor Gold later. (med, dairy)*

*I've never used it as part of any kind of weight gain or anything like that. . . .For prevention, I give the entire cow herd 5,600 grams per ton of Aureomycin per ton of mineral during the months of the fly season which basically is from May until November. (medium, beef cattle operator, needing to prevent severe blood disease, anaplasmosis.)*

## Ohio livestock ten-year production trends

In general, longitudinal data on the level of Ohio and US cattle production suggest that there were no obvious negative impacts on overall production output associated with the implementation of GFI #213 and the revised Veterinary Feed Directive (VFD). The vertical lines in Fig 5

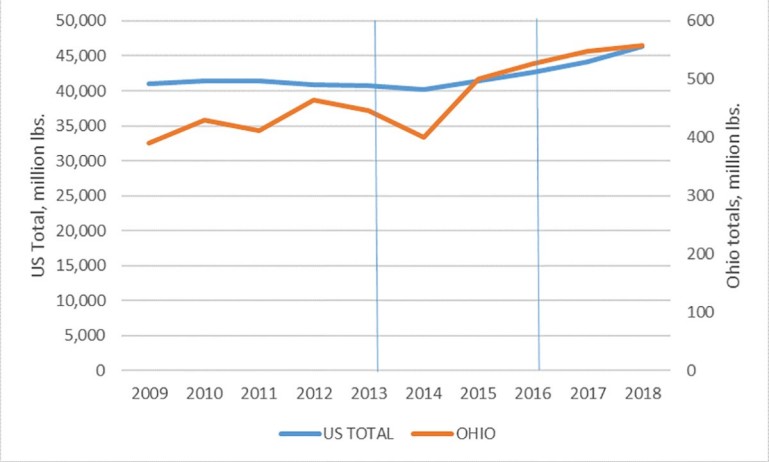

**Fig 5. Cattle production, measured in pounds, 2009–2018.** (Source, USDA). Vertical lines indicate the beginning of voluntary (left) and mandatory (right) compliance with the FDA's GFI #213, restricting use of antibiotics in livestock.

represent the years when voluntary (left line) and mandatory (right line) compliance were initiated, respectively. Both US and Ohio cattle production have been steadily increasing since the implementation of GFI #213 (Fig 5), suggesting that production by Ohio cattle operations have not been negatively impacted by the new FDA regulations.

These data are consistent with our cattle producer interviews. The overwhelming majority of interviewees reported "no impact" on production as the result of the VFD. During the interviews, some farm operators reported challenges in 2019 due to political and/or economic reasons (e.g. China tariffs, changing economies of scale); however, none listed the VFD as a negative driver of production.

## Discussion

The most surprising and important finding of the interviews may be that Ohio cattle producers and veterinarians generally experienced minimal difficulty in transitioning to the new rules implemented under GFI #213. Mandates mean nothing if they are not able to change behavior at the farm level. The transition to the new rules appears to have gone smoothly, with the FDA reporting a high rate of compliance [27]. Based on our interviews, it is clear that many Ohio cattle producers were relatively unaffected by the VFD, due to either lack of prior use of fed antibiotics or the fact that the operation was already trending toward less antibiotic use. With respect to the former, 44% reported that they had not previously used fed antibiotics. It may be worth noting that over one-half of Ohio beef operations and one-third of Ohio dairy operations consist of small herds numbering fewer than ten head [22]. Over two-thirds of surveyed operators from these farm size categories reported using no fed antibiotics prior to the implementation of the feed directive; predictably, they reported minimal to no impact.

Several producers, typically medium to large operations, reported that they were already trending toward minimizing antibiotic use and increasing vaccination rates. It is unclear, however, whether these trends were triggered, at least in part, in anticipation of GFI #213. Many in the agricultural industry, especially larger operations and veterinarians, were aware of the upcoming mandate. Our interviews occurred from May 2019 to January 2020. Therefore, the industry had already experienced three years of voluntary compliance (2014–2016) plus 2 ½ to 3 years of mandatory compliance. According to the FDA summary reports, sales of MIA for livestock decreased nationally for the first time (by 14%) in 2016 (i.e. the last the year prior to implementation of GFI #213). This suggests that operational practices on many farms were already being modified in anticipation of the feed directive. Additionally, a few producers mentioned that they had chosen to eliminate antibiotics from some herds to benefit financially from consumer preference.

According the 2018 FDA Summary report, nation-wide consumption of antibiotics (measured in Kg) by cattle decreased 35% in 2017, the first year of GFI #213 implementation. It is interesting to note that Ohio veterinarians perceived a more substantial decrease in both fed and total antibiotic use than their producer counterparts. In fact, most of Ohio cattle producers using fed antibiotics reported "little" to "no change" in the total use of antibiotics with their herds. One explanation may be that veterinarians may be more cognizant of actual dosage, while producers who shifted to more injectable antibiotics may be reporting usage by increased frequency. Our interview instrument did not clearly delineate antibiotic usage in terms of amount versus frequency. A second possible explanation is that national averages may be more representative of large feedlot operations in the western US and may not be as reflective of the typical Ohio cattle operation.

The impressions of the interviewed Ohio cattle producers showed interesting similarities and differences with the Tennessee (TN) cattle producers surveyed by Ekakoro et al [19].

Broadly speaking, the TN cattle producers expressed much more negative perceptions of the VFD (70% of those surveyed expressed negative attitudes), often citing the VFD as unnecessary "top-down" control. The TN producers also expressed concerns that the VFD would negatively impact production, herd health, and the economics of the cattle industry. Ohio producers, by contrast, consistently reported "no change" in terms of herd health and production, and "little" to "no change" with respect to profitability (due to the VFD). The perceived lack of impact on production was supported by longitudinal data on both Ohio and national level production trends. Perceptions of the VFD by Ohio producers ranged from continued minor frustrations ("we were grumbling, about this, just this morning'") to more neutral ("It's OK; it's just a hassle") to positive ("It was absolutely needed.") Although the Ohio cattle producers were generally more positive about the feed directive, many expressed feelings of frustration that agriculture was over-regulated in general and unfairly blamed by society for many public woes.

One explanation for these differences in perceptions that the Tennessee interviews began in mid-2017, in the middle of the first year of implementation of GFI #213. In fact, the study noted that many of its participants were either "unaware" or "only slightly aware" of the new mandate. After 2 ½ - 3 years of operating under the new FDA regulations, it is likely that Ohio cattle producers had adjusted (if needed), and for the most part, adjusted well.

Consistent between both studies, dairy operators generally seemed less convinced of the usefulness of VFD, but also more prepared for it due to existing VCPRs and more regular interactions with veterinarians. Additionally, both studies reported the perception that smaller cattle operations would have greater difficulty with compliance, primarily due to lack of access to veterinarians, especially in more remote areas.

In order to compensate for the restrictions on MIA, both groups of cattle producers reported some increase in the use of vaccines and nutritional supplements, though less of the latter in Ohio [20]. These strategies may be common and increasing across livestock sectors. A recent survey of swine veterinarians reported that the strategies their clients employed to compensate for the restrictions on antibiotics were, in descending order: increased vaccinations, increased non-antibiotic feed additives, modified biosecurity, and modified nutrition [21]. Also, Duttlinger reported that supplementation of a single amino acid (L-gluttamine) in piglet feed enhanced both production and wellness [28].

A common theme in the Ohio interviews, among many producers and all veterinarians, is an increased quantity and quality of veterinarian-client interactions. This is consistent with Rademacher [21], where swine veterinarians reported an increase in discussions with their producer clients regarding antibiotic use and alternatives.

Perhaps the most striking impact of the VFD on Ohio stakeholders, however, was the increased amount of record-keeping and paperwork. The great majority of surveyed Ohio cattle farmers who previously used fed antibiotics as well as 100% of the veterinarians reported this impact. This increase was also reported in the other VFD studies cited. Ironically, this impact—which was widely reported as a "nuisance'—may be largely responsible for the decrease and more judicious use of antibiotics. According to industry leaders, the substantial decrease in antibiotic consumption by chickens was prodded by the paperwork mandated by the VFD [29]. This record keeping has allowed producers and veterinarians to evaluate antibiotic usage and efficaciousness more readily in given situations.

Many—though not all—livestock producers and veterinarians reported a shift in philosophy from using antibiotics as the "first line of defense." Our Ohio interviews suggested diverse viewpoints on the prophylactic or metaphylactic use of antibiotics in agriculture in the aftermath of the VFD. Some maintained that preventative measures can reduce or even eliminate the need for preventative antibiotic use in herds while others maintained that in certain

situations this use is warranted or even essential. For instance, a pasture-based cattle producer reported he was able to virtually eliminate use of all antibiotics by taking great care to manage the stress level in his herds. In contrast, a cattle producer fighting *anaplasmosis* reported the need to use preventative antibiotics during fly season to simply preserve his herd. One veterinarian maintained that metaphylaxis has been demonstrated to be effective in large feedlots while another suggested the need was minimal if animals entered and existed as a cohort, sanitizing in between arrivals. Many informants recognized the societal pressure to reduce or eliminate the use of preventative antibiotics in general.

## Recommendations for further research

The goal of the FDA in implementing the revised VFD and GFI #213 was to promote a decreased and a more judicious use of medically important antibiotics in livestock production. Federal statistics suggest that antibiotic consumption in livestock nationwide has significantly decreased since the implementation of GFI #213 [10]. Similar statistics are not available at the state-level. To fill this gap, future farm-level research in Ohio would benefit by tracking overall volumes of antibiotic use over time on a representative sample of farms. It would also be interesting to replicate this work in other states to see if the patterns found in Ohio are similar or different to the experiences in regions with different types of cattle enterprises.

Producers themselves have voiced a greater need for more scientific research, especially in the area of genetics, specifically citing both the need for more disease-resistant livestock and continued research into greater feed conversion. Finally, in January 2022, the FDA plans to issue another rule (GFI #263) which will move the remaining OTC antibiotics, including injectables, under the supervision of veterinarians. A similar post-implementation farm-level study could help identify how this next policy change impacts different types of cattle producers.

## Study limitations

To our knowledge, this is the first study to investigate the impact of the revised VFD on Ohio livestock operations; however, our work has several limitations. The farms included in our interviews were not randomly selected, although we did try to achieve a theoretical quota sample to get a range of farm types and sizes. Our producer interviews focused only on cattle operations, and did not include poultry and swine production. Finally, our investigation of local level impacts of the VFD were restricted to Ohio operations and the experiences may differ in other states, particularly those with a greater predominance of large-scale cattle feedlot operations.

## Conclusions

Our findings are encouraging. GFI #213 appears to have accomplished its goal of promoting more judicious use of antibiotics in livestock production. The revised VFD was reported to be not onerous enough to prevent compliance, but inconvenient enough to incentivize the reduced use of fed antibiotics. Thus, the new mandates appear to be working in terms of decreasing the consumption of antibiotics by livestock among operations previously using fed antibiotics without significant adverse effects on production. Additionally, they bring the agricultural administration of medically important antibiotics more in-line with how these same antibiotics are administered in the human sector, i.e. under the supervision of a health professional.

## Supporting information

**S1 File. Producer interview questions, phase 1.**
(DOCX)

**S2 File. Producer interview questions, phase 2.**
(DOCX)

**S3 File. Veterinarian interview questions, phase 2.**
(DOCX)

**S4 File. Deidentified dataset.**
(XLSX)

## Author Contributions

**Conceptualization:** Mary Ellen Dillon, Douglas Jackson-Smith.

**Data curation:** Mary Ellen Dillon, Douglas Jackson-Smith.

**Formal analysis:** Mary Ellen Dillon, Douglas Jackson-Smith.

**Funding acquisition:** Douglas Jackson-Smith.

**Investigation:** Mary Ellen Dillon, Douglas Jackson-Smith.

**Methodology:** Mary Ellen Dillon, Douglas Jackson-Smith.

**Supervision:** Douglas Jackson-Smith.

**Writing – original draft:** Mary Ellen Dillon.

**Writing – review & editing:** Douglas Jackson-Smith.

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
