## [Decision Letter · Decision Letter 0]

25 Mar 2021

PONE-D-21-05230

Impact of the Veterinary Feed Directive on Ohio cattle operations

PLOS ONE

Dear Dr. Dillon,

Thank you for submitting your manuscript to PLOS ONE. After careful consideration, we feel that it has merit but does not fully meet PLOS ONE’s publication criteria as it currently stands. Therefore, we invite you to submit a revised version of the manuscript that addresses the points raised during the review process.

Both expert reviewers are supportive of this manuscript being considered further for publication, and I concur with their view. Nevertheless, they have highlighted some important points that prevent the manuscript from acceptance as it stands. Please revise your manuscript attending to the reviewers' comments, particularly regarding the detailed reporting of data collection and trying to convey the messages more succinctly.

We look forward to receiving your revised manuscript.

Kind regards,

Angel Abuelo, DVM, MRes, MSc, PhD, DABVP (Dairy), DECBHM

Academic Editor

PLOS ONE

Journal Requirements:

1. Please ensure that your manuscript meets PLOS ONE's style requirements, including those for file naming. The PLOS ONE style templates can be found athttps://journals.plos.org/plosone/s/file?id=wjVg/PLOSOne_formatting_sample_main_body.pdf andhttps://journals.plos.org/plosone/s/file?id=ba62/PLOSOne_formatting_sample_title_authors_affiliations.pdf

'D.J. received partial funding for the research reported here from a seed grant from The

Ohio State University’s Initiative for Food and AgriCultural Transformation (InFACT), a

Discovery Themes program (discovery.osu.edu/infact). The funder played no role in

the study design, data collection or analysis, decision to publish, or preparation of the

manuscript.'

Additional Editor Comments (if provided):

Reviewers' comments:

Reviewer's Responses to Questions

**Comments to the Author**

1. Is the manuscript technically sound, and do the data support the conclusions?

Reviewer #1: Partly

Reviewer #2: Yes

2. Has the statistical analysis been performed appropriately and rigorously? 

Reviewer #1: No

Reviewer #2: N/A

3. Have the authors made all data underlying the findings in their manuscript fully available?

Reviewer #1: Yes

Reviewer #2: No

4. Is the manuscript presented in an intelligible fashion and written in standard English?

Reviewer #1: Yes

Reviewer #2: Yes

5. Review Comments to the Author

Reviewer #1: The paper was well-written and contains good information. The use of quotes is good in order to flesh out views and actions. But I have 3 major problems with it; 1) data collection description, 2) data use and 3) conclusions.

1. Data collection: The authors do not cite how many different interviewers were involved and whether there was difference (ie. number of questions answered) by interviewer. My difficulty is in understanding if the two phases can be directly added to each other.

In the description of Table 2 you state that "Not all questions were answered by all participants since some items were only included in Ph. II", however, "# of Vet interactions" with n=20 and "Amt of paperwork" n=16 were included in both Phase surveys. So, why the low “n”? In a verbally administered questionnaire, I wouldn’t expect to see such a low “n”.

In regard to the survey of 8 vets total, it would be good (I believe necessary) to include the "n" for each question in Fig. 4.

Based on these comments, I'm not sure that my response on question 3 above is accurate.

2. Data use: The two (Ph. I & II) surveys are different and only the data from common questions can be combined. But it seems that you do more than that. For instance, Line:297-98 states that 22 of 54 had not been using fed antibiotics, but I don’t see that specific question on the Phase I interview questionnaire. In fact, you cite that issue in the description under Table 2, but somehow have the data on all farms. Is it inferred from question 1 (under Antibiotic use) about ever using antibiotics? I don’t believe it can be inferred. This becomes especially important given that 80% of the dairy interviews were in Phase I. Yet, Line:303-04 states that 11 of the 15 reported pre-VFD use of fed antibiotics. What question did that information come from?

On Line:311-12 you state that there was a significant decrease in the use of fed Ab for dairy calves (I can believe that). And on Lines:388-89 you refer to beef operations that previously used fed Ab saying "the most common change included reduced use of fed Ab". Yet on Lines:725-25 you state that "the great majority of OH cattle producers using fed An reported 'little' to 'no change' in cattle consumption of Ab." These seem contradictory.

3. Conclusions: in your conclusions, you do much more than conclude what the data reveal and editorialize to a great extent about how that is a good thing. I am not arguing whether it is good or bad, but the data don't tell you that.

You have some great points that you make including Lines: 806-810, but this paper seems hamstrung by inconsistent data collection.

Reviewer #2: Dear authors,

I have now reviewed your manuscript. From my perspective your manuscript contributes significantly to the creation of knowledge in this area of study, has been conducted using scientific and robust methods, and is very well-written. The outcomes of your study can support policy development and implementation.

I only have some minor comments for you to consider.

Line 220: Phase II focused... check this sentence, I think 'For' should be deleted

Line 222: It seems you have used snowballing techniques to identify potential study participants. It might be worth mentioning the technique used.

Line 248: It would be good if you can provide additional details on the methodology of the qualitative data analysis in addition to indicating that themes and patterns were identified.

Line 374: Please provide an example of what you mean by 'aggressive preventative measures'

Line 488: Was there any other difference between veterinarians in addition to the proportion of cattle producers served and the years from graduation? I think some more details on participant veterinarians would be good so you provide better understanding of the diversity and representativeness of this group.

Another general comment is that the manuscript is quite long. You provide a lot of quotes, which is great but might be too many. I recommend you to review and identify opportunities for reduction of the number of quotes for each of the themes.

6. PLOS authors have the option to publish the peer review history of their article (what does this mean?). If published, this will include your full peer review and any attached files.

Reviewer #1: No

Reviewer #2: No

---

## [Author Response · Author response to Decision Letter 0]

19 Jun 2021

RESPONSE TO REVIEWERS PONE-D-21-05230 

Impact of the Veterinary Feed Directive on Ohio cattle operations

Responses to Reviewer Comments to the Author

Below we have inserted our responses to each reviewer comment in bulleted italicized text.

Reviewer #1:

The paper was well-written and contains good information. The use of quotes is good in order to flesh out views and actions. But I have 3 major problems with it; 1) data collection description, 2) data use and 3) conclusions.

Thank you for the detailed review of our manuscript. In reviewing the submission, we discovered several issues which directly contributed to the lack of clarity in our methods and results sections. Below we review each issue and how we addressed it in the revision.

1. Data collection: The authors do not cite how many different interviewers were involved and whether there was difference (ie. number of questions answered) by interviewer. My difficulty is in understanding if the two phases can be directly added to each other.

● We edited the manuscript to clarify how the two phases are comparable and can be combined for nearly all of the topics covered in the paper. Specifically:

○ All phase I interviews were done with pairs of interviewers (one to ask questions, the other to take notes); all phase II interviews were conducted by the lead author. At least one of the paper’s authors was present at 29/35 of the phase I interviews and a third lead interviewer was present at the remaining 6. The lead author did the second phase interviews alone. All interviews were recorded (with the participants’ permission.) All recordings and transcripts were available to both authors.

○ The same interview instrument was used by all field interviewers within each phase, and all questions were asked in a similar manner. In each case, consistent with a semi-structured interview method, the interviewers typically asked probing questions to encourage respondents to elaborate or clarify their answers when they were ambiguous. Interview transcripts were used to confirm that questions were asked consistently and complete information was obtained wherever possible.

○ Phase II interviews were shorter than phase I (and focused mainly on the VFD topic), but detail about the VFD impacts was elaborated by adding a couple of new open-ended prompts, and adding two new items to the Likert-scale table used to capture changes since VFD from Phase I. These two-items included: The Effect of the VFD on the amount of fed antibiotics and the amount of paperwork. Both new Likert items covered topics that arose in the open-ended question on VFD impacts from phase I, and (as we discuss below), we were able to derive values for the change in fed antibiotics item for most phase I respondents using their detailed narrative answers. We did not attempt to derive answers for phase I respondents on the paperwork item since that was not systematically discussed.

○ Overall, we were very pleased with the consistency of the interviews, both in how the questions were asked and the number of questions answered. We acknowledge that some phase I questions on VFD and AB use were slightly rephrased, reordered, or elaborated in phase II (see supplemental files). However, we believe that the core information from each phase is comparable on most items.

In the description of Table 2 you state that "Not all questions were answered by all participants since some items were only included in Ph. II", however, "# of Vet interactions" with n=20 and "Amt of paperwork" n=16 were included in both Phase surveys. So, why the low “n”? In a verbally administered questionnaire, I wouldn’t expect to see such a low “n”.

• Thank you for pointing out this issue. 

• Upon further review of our materials, we realized that we had submitted an inaccurate copy of the phase I instrument with the original supplemental files (specifically – we accidentally uploaded an early draft version). The final version of the instrument that was actually used in Phase I did include a question about veterinarian interactions but did not include a question about paperwork. In addition, for related reasons, we realized that the vet interactions item was not systematically coded from phase I interviews when field data were entered to a shared Access database. 

• We went back to the raw data (field notes and transcripts) and updated the quantitative dataset to capture these additional cases. In the process, we also fixed a few minor data gaps and confirmed instances where data were missing. Missing data reflected instances where either the farmer elect to not answer a specific question or where their answers were ambiguous, or (as in 1 phase II case) where the farm did not exist prior to VFD).

• Separately from the paperwork and vet interaction items, we also used this opportunity to systematically assess whether or not we could derive answers to the ‘fed antibiotic’ impact scale item for phase I farms. Simply put – although we did not add the topic as a formal item to the interview instrument until Phase II, in nearly every case the qualitative and open-ended responses to the VFD question block generated sufficient information to identify cases where fed AB use was unchanged (e.g., where no fed ABs had ever been used), and where fed AB use dropped significantly (e.g., where they pointed out how they used to feed ABs prior to the VFD, but no longer did so). The updated table 1 (and uploaded deidentified dataset) provides more complete information about the changes in fed antibiotics for both phase I and phase II farms.

• The revised manuscript now reports complete and accurate information in Table 1 about the number of cases where farmers reported how VFD impacted them along each of the various metrics. We have also replaced Fig 2 with a new figure that only shows changes made on farms that had previously fed antibiotics prior to the VFD. We feel this shows more clearly how common different changes were among the population of farms most directly impacted by the new rule. 

• A complete and validated version of the dataset is now included as a supplemental file with our submission. We have also updated the versions of the interview instruments to ensure that the correct phase I instrument can be seen. We apologize for our errors in the original submission, but appreciate the chance to review and reanalyze all of our data for this resubmission.

In regard to the survey of 8 vets total, it would be good (I believe necessary) to include the "n" for each question in Fig. 4.

• We have complete information for all vets, so felt that it was unnecessary to include separate Ns for each item. We have clarified this in the text.

 2. Data use: The two (Ph. I & II) surveys are different and only the data from common questions can be combined. But it seems that you do more than that. For instance, Line:297-98 states that 22 of 54 had not been using fed antibiotics, but I don’t see that specific question on the Phase I interview questionnaire. In fact, you cite that issue in the description under Table 2, but somehow have the data on all farms. Is it inferred from question 1 (under Antibiotic use) about ever using antibiotics? I don’t believe it can be inferred. This becomes especially important given that 80% of the dairy interviews were in Phase I. Yet, Line:303-04 states that 11 of the 15 reported pre-VFD use of fed antibiotics. What question did that information come from?

• We appreciate the concern and request for more information. 

• The interview instrument in phase I asked farmers if they ever used antibiotics with their cattle and asked them to describe their overall approach and provide several examples of recent instances in which antibiotics were administered to their cattle. Separately, we asked a broad open-ended question about how the VFD affected their livestock operation, and another about the biggest ‘challenges’ to their farm from VFD. Additionally, we prompted farmers who didn’t describe a change following VFD to find out if they had ever fed ABs (not obvious on the instrument, but common in our interview transcripts). In nearly all cases, we were able to determine from answers to these open-ended questions whether or not they were feeding ABs prior to the VFD (and if so, whether that continued after the VFD). This information is provided in the new supplemental quantitative dataset. When we revised the instrument for phase II, we decided to make this question more explicit. It is true that the results may be slightly different given the different interview question phrasing, but believe that we captured comparable information from both phases of the study. 

• As noted above, in addressing your concerns, we decided to revisit the transcripts and field notes and determined that we could not only determine which operators had previously used fed antibiotics but also the amount of change caused by the revised VFD. We have adjusted our Table 2 to reflect this and included complete information in the new quantitative data supplemental file.

• We have edited the text to make clear that there could be some undercounting or measurement error on the prevalence of pre-VFD fed AB use for phase I farms. 

On Line: 311-12 you state that there was a significant decrease in the use of fed Ab for dairy calves (I can believe that). And on Lines:388-89 you refer to beef operations that previously used fed Ab saying "the most common change included reduced use of fed Ab". Yet on Lines:725-25 you state that "the great majority of OH cattle producers using fed Abs reported 'little' to 'no change' in cattle consumption of Ab." These seem contradictory.

• Most dairy farms had fed ABs to calves prior to the VFD. When both dairy and beef producers had previously fed ABs (prior to the VFD), their use of fed ABs generally declined (though not always, particularly when their veterinarian was able to provide a VFD that permitted continued use). However, few farmers in our sample suggested that overall AB use declined in their herds. As for line 725, you are correct, that line is confusing as written. We should have used the word “use” vs. “consumption” which implies “fed.”. We have edited to clarify to show how the two claims are not contradictory.

3. Conclusions: in your conclusions, you do much more than conclude what the data reveal and editorialize to a great extent about how that is a good thing. I am not arguing whether it is good or bad, but the data don't tell you that. 

• Thank you. We have significantly revised this section to minimize the editorial comment and restricted our conclusions to things directly supported by the data.

You have some great points that you make including Lines: 806-810, but this paper seems hamstrung by inconsistent data collection.

• We appreciate the questions about the differences between Phase I and II of the project. We believe the data are quite consistent, and hopefully have clarified this in the revised manuscript and addressed it to your satisfaction in our response comments above.

 

Reviewer #2: 

I have now reviewed your manuscript. From my perspective your manuscript contributes significantly to the creation of knowledge in this area of study, has been conducted using scientific and robust methods, and is very well-written. The outcomes of your study can support policy development and implementation.

I only have some minor comments for you to consider.

Line 220: Phase II focused... check this sentence, I think 'For' should be deleted

• Thank you. We made this edit.

Line 222: It seems you have used snowballing techniques to identify potential study participants. It might be worth mentioning the technique used.

• Good point. We have edited the methods to acknowledge our use of this approach.

Line 248: It would be good if you can provide additional details on the methodology of the qualitative data analysis in addition to indicating that themes and patterns were identified.

• We have elaborated on the approach to qualitative data analysis methods in the manuscript. For this paper, the qualitative analysis was fairly straightforward – e.g., both authors were able to identify text in the transcripts that represented farmer responses to each interview question. We then assigned codes to that text indicating the question to which it applied, and then looked across the set of answers within each question block to identify some subthemes or emergent codes that were apparent in the overall set of answers to the same question or sets of questions. 

• Both authors did this independently and compared coding decisions to identify areas of potential disagreement, then refined the criteria for classification to ensure reliability and validity. 

• For example, we were able to agree on a set of codes to capture different levels of difficulty in response to the question ‘how difficult was it for your operation to comply with the VFD.” We also used multiple questions (particularly in phase I) to create a codes to identify whether or not they had ever fed antibiotics to cattle and how that changed in response to the VFD. Contributing questions to creating those codes included asking about their ‘overall approach to antibiotic use,’ whether ‘over the last 12 months have you used any antibiotics in your dairy or beef herd’, ‘how did (the VFD) affect your livestock operation’ and ‘what was the greatest challenge for your operation in complying with the VFD.’ 

• For the present manuscript, the most common answers reported in the paper were in response to the questions reflecting the types of changes that they had to make and the overall impact of the VFD on their operations.

Line 374: Please provide an example of what you mean by 'aggressive preventative measures'

• We replaced the word ‘aggressive’ with ‘strong’ and have provided a few examples of preventative measures in the revised text. 

Line 488: Was there any other difference between veterinarians in addition to the proportion of cattle producers served and the years from graduation? I think some more details on participant veterinarians would be good so you provide better understanding of the diversity and representativeness of this group.

• We didn’t ask for much more detailed background or demographic information from the veterinarian informants. We did attempt to talk to veterinarians from across the diverse study counties to account for regional variability.

Another general comment is that the manuscript is quite long. You provide a lot of quotes, which is great but might be too many. I recommend you to review and identify opportunities for reduction of the number of quotes for each of the themes.

• Thank you for this observation. We have reorganized the results section and deleted many quotes where they did not present additional information to help address this length issue. We also cut most the sections on ‘Emergent Themes’ and abbreviated some of the discussion and conclusions to remove content that was not essential to answering our core research questions.

---

## [Decision Letter · Decision Letter 1]

27 Jul 2021

Impact of the Veterinary Feed Directive on Ohio cattle operations

PONE-D-21-05230R1

Dear Dr. Dillon,

We’re pleased to inform you that your manuscript has been judged scientifically suitable for publication and will be formally accepted for publication once it meets all outstanding technical requirements.

Kind regards,

Angel Abuelo, DVM, MRes, MSc, PhD, DABVP (Dairy), DECBHM

Academic Editor

PLOS ONE

Additional Editor Comments (optional):

Reviewers' comments:

Reviewer's Responses to Questions

**Comments to the Author**

1. If the authors have adequately addressed your comments raised in a previous round of review and you feel that this manuscript is now acceptable for publication, you may indicate that here to bypass the “Comments to the Author” section, enter your conflict of interest statement in the “Confidential to Editor” section, and submit your "Accept" recommendation.

Reviewer #1: All comments have been addressed

Reviewer #2: All comments have been addressed

2. Is the manuscript technically sound, and do the data support the conclusions?

Reviewer #1: Yes

Reviewer #2: Yes

3. Has the statistical analysis been performed appropriately and rigorously? 

Reviewer #1: Yes

Reviewer #2: N/A

4. Have the authors made all data underlying the findings in their manuscript fully available?

Reviewer #1: Yes

Reviewer #2: Yes

5. Is the manuscript presented in an intelligible fashion and written in standard English?

Reviewer #1: Yes

Reviewer #2: Yes

6. Review Comments to the Author

Reviewer #1: This version is significantly improved over the original submission and is acceptable for publication. Thank you for the further work on the manuscript. I will call to your attention a few lines that should be looked at to make sure they read right. These are for your consideration.

Line 80: ". . . administration through food and water."

Line 95: ". . . among US meat- and milk-producing sectors."

Lines 268 and 270: phase I is capitalized on lines 261 and 262, I think that for the sake of consistency, Phase should be capitalized on lines 268 and 270.

Line 390: I believe it should read either ". . . they never used fed antibiotics." or ". . . they never fed antibiotics."

Reviewer #2: The authors have addressed appropriately all the comments I had from the first revision and the manuscript has improved significantly

7. PLOS authors have the option to publish the peer review history of their article (what does this mean?). If published, this will include your full peer review and any attached files.

Reviewer #1: **Yes: **Phillip Durst

Reviewer #2: No

---

## [Editor Report · Acceptance letter]

30 Jul 2021

PONE-D-21-05230R1 

Impact of the Veterinary Feed Directive on Ohio cattle operations 

Dear Dr. Dillon:

I'm pleased to inform you that your manuscript has been deemed suitable for publication in PLOS ONE. Congratulations! Your manuscript is now with our production department. 

Kind regards, 

on behalf of

Dr. Angel Abuelo 

Academic Editor

PLOS ONE